# DYRK1A Up-Regulation Specifically Impairs a Presynaptic Form of Long-Term Potentiation

**DOI:** 10.3390/life15020149

**Published:** 2025-01-22

**Authors:** Aude-Marie Lepagnol-Bestel, Simon Haziza, Julia Viard, Paul A. Salin, Arnaud Duchon, Yann Herault, Michel Simonneau

**Affiliations:** 1Centre Psychiatrie & Neurosciences, INSERM U894, 75014 Paris, France; aude-marie.lepagnol-bestel@univ-reims.fr (A.-M.L.-B.); sihaziza@stanford.edu (S.H.); julia.viard@gmail.com (J.V.); 2Centre National de la Recherche Scientifique, Université Paris-Saclay, CentraleSupélec, École Normale Supérieure Paris-Saclay, LuMIn, 91190 Gif-sur-Yvette, France; 3Centre de Recherche en Neuroscience de Lyon CRNL (INSERM U1028), Université Claude-Bernard Lyon 1, 69100 Lyon, France; paul.salin@sommeil.univ-lyon1.fr; 4Institut de Génétique et de Biologie Moléculaire et Cellulaire (IGBMC), Université de Strasbourg, CNRS UMR7104, INSERM, U964, 67404 Illkirch, France; duchon@igbmc.fr (A.D.); herault@igbmc.fr (Y.H.); 5Phenomin, Institut Clinique de la Souris (ICS), GIE CERBM, CNRS, INSERM, Université de Strasbourg, 1 rue Laurent Fries, 67404 Illkirch, France; 6Département d’Enseignement et de Recherche en Biologie, École Normale Supérieure Paris-Saclay, 91190 Gif-sur-Yvette, France

**Keywords:** Down syndrome, synapse, long-term potentiation, NMDA-independent LTP, pre-synaptic mechanisms, epigenetics

## Abstract

Chromosome 21 DYRK1A kinase is associated with a variety of neuronal diseases including Down syndrome. However, the functional impact of this kinase at the synapse level remains unclear. We studied a mouse model that incorporated YAC 152F7 (570 kb), encoding six chromosome 21 genes including *DYRK1A*. The 152F7 mice displayed learning difficulties but their N-methyl-D-aspartate (NMDA)-dependent synaptic long-term potentiation is indistinguishable from non-transgenic animals. We have demonstrated that a presynaptic form of NMDA-independent long-term potentiation (LTP) at the hippocampal mossy fiber was impaired in the 152F7 animals. To obtain insights into the molecular mechanisms involved in such synaptic changes, we analyzed the Dyrk1a interactions with chromatin remodelers. We found that the number of DYRK1A-EP300 and DYRK1A-CREBPP increased in 152F7 mice. Moreover, we observed a transcriptional decrease in genes encoding presynaptic proteins involved in glutamate vesicle exocytosis, namely Rims1, Munc13-1, Syn2 and Rab3A.To refine our findings, we used a mouse BAC 189N3 (152 kb) line that only triplicates the gene *Dyrk1a*. Again, we found that this NMDA-independent form of LTP is impaired in this mouse line. Altogether, our results demonstrate that *Dyrk1a* up-regulation is sufficient to specifically inhibit the NMDA-independent form of LTP and suggest that this inhibition is linked to chromatin changes that deregulate genes encoding proteins involved in glutamate synaptic release.

## 1. Introduction

Down syndrome (DS), a human genetic disorder, is the most common form of intellectual disability (ID), affecting 1/1000 births. DS is caused by the presence of a third copy of up to 221 coding genes from *Homo sapiens* autosome 21 (Hsa21) [1] (see https://jul2023.archive.ensembl.org/Homo_sapiens/Location/Chromosome?r=21 (accessed on 21 December 2024)).

Despite a broad spectrum of clinical symptoms, virtually all people affected by DS develop Alzheimer’s disease pathology by 40 years of age, with intellectual deficits that impair learning and memory [1,2,3,4]. However, a detailed understanding of the precise contribution of each Hsa21 gene to the cognitive impairment found in DS patients is still lacking.

The chromosome 21 DYRK1A (dual-specificity tyrosine phosphorylated and regulated kinase 1A) gene is a major player in DS, and its overexpression impacts the synaptic plasticity within the hippocampus and the prefrontal cortex [5,6,7].

Converging evidence suggests that DYRK1A might be critical for learning and memory processes [8,9]. However, a clear understanding of the regulatory pathways impaired by a trisomy of DYRK1A is still lacking. Synaptic long-term potentiation (LTP) is involved in learning and memory [10]. Two main types of LTP have been described as follows: an NMDA-dependent LTP and an NMDA-independent LTP [11]. We investigated whether LTP is impaired at the dentate gyrus granule cells to CA3 pyramidal cell synapse (ie Mossy Fiber LTP, MF LTP), which is known to be a NMDA-independent presynaptic LTP and linked to presynaptic proteins [12,13].

Here, we used two DS mouse models. The first model is a human YAC mouse model (Tg(CEPHY152F7)12Hgc, noted here as the 152F7 line) (https://www.informatics.jax.org/ (accessed on 21 December 2024)) that displays a ~570 kb human genomic region surrounding the *DYRK1A* gene and includes five other genes [14]. The 152F7 mouse model incorporates a 570 kb fragment of Hsa21 with six protein-coding genes including DYRK1A (Appendix A). One of the six genes, *DSCR9*, is a primate-specific gene [15,16]. The syntenic region in the mouse chromosome 16 genome is more condensed with ~350 kb (Appendix A) instead of ~570 kb in Hsa21. The organization of genes is similar between the human and mouse genome with the following sequence from 3′ to 5′: *Ripply3*, *Pigp*, *TTC3*, *Vps26c* and *Dyrk1a*.

The second is a *Dyrk1A* BAC model (Tg(Dyrk1a)189N3Yah, noted here as the 189N3 line) that gives a triplication of ~152 kb mouse Dyrk1a locus [6]. Interestingly, several studies reported that these lines display behavioral defects in episodic-like forms of memory [14,17]. Indeed, both 152F7 and 189N3 mouse models showed an impairment in long-term memory tested using the Morris water maze [14,17] and novel object recognition task [18].These results indicate that both the “where” and the “what” pathways of episodic memory [19] are impacted in these models.

Interestingly, 152F7 mice display learning defects but their NMDA-dependent synaptic long-term potentiation (LTP) in the hippocampus is indistinguishable from non-transgenic animals [6].

In this manuscript, our aim was (i) to demonstrate a relationship between the cognitive defects found in these two mouse models (152F7 line and 189N3 line) and a possible functional synaptic defect in these mouse lines and (ii) to obtain insights into the possible molecular mechanisms involved.

Therefore, using extracellular field recordings in hippocampal slices, we found that the MF LTP is specifically impaired in these two mouse lines.

We previously showed that DYRK1A interacts with the REST/NRSF-SWI/SNF chromatin remodeling complex to deregulate gene clusters involved in the neuronal phenotypic traits of Down syndrome [20]. By performing exome sequencing and mass spectroscopy in these mouse models, we recently uncovered two deregulated repertoires associated with the chromatin and synaptic pathways [21].

To obtain insights into the molecular mechanisms involved in such synaptic changes, we analyzed DYRK1A interactions with chromatin remodelers. Here, we found that the number of DYRK1A-EP300 and DYRKA-CREBBP increased in 152F7 mice, using proximity ligation assay (PLA) technology. Moreover, we observed a transcriptional decrease in genes encoding presynaptic proteins involved in glutamate vesicle exocytosis, namely, *Rims1, Munc13–1, Syn2* and *Rab3a*.

Altogether, these results reveal the previously unobserved impact of Hsa21 DYRK1A on synaptic plasticity with possible consequences for a variety of neuropsychiatric diseases as diverse as intellectual disability, late-onset-Alzheimer disease and autism spectrum disorders.

## 2. Materials and Methods

### 2.1. Animals and Genotyping

We used wild-type mice of the OF1 strain for neuronal primary culture, as well as mice of the C57BL6, 189N3 (Tg(Dyrk1a)189N3Yah (10.1016/j.nbd.2012.01.007)) [6] and 152F7 (Tg(CEPHY152F7)12Hgc, 10.1006/geno.1995.1073) transgenic lines [14]. Genotypes were determined using genomic DNA extracted from skeletal muscle fragments and the PCR protocol and primers as previously described [20].

### 2.2. Quantitative In Situ Hybridization (ISH)

For the Rim1 in situ hybridization, a 1124 bp Rim1 PCR product (covering nucleotides 376 to 1499 of the XM_129709 cDNA sequence) was inserted into a pCRII-TOPO cloning vector). XhoI- or HindIII-linearized Rim1 inserts were used to generate antisense or sense [α35S]-rUTP (800 Ci/mmol, Amersham)-labeled transcripts, using the P1460 riboprobe in vitro transcription systems (Promega), according to the manufacturer’s instructions. Paraffin-embedded coronal 15 µm sections of P21 mouse brains, including hippocampal structures, were selected for ISH. Sections were hybridized with a radiolabeled RNA probe (diluted to 105 cpm/µL in 50% formamide hybridization buffer) in 50% formamide at 50 °C. Sections were successively washed in 50% formamide, 2 × SSC, 10 mM DTT at 65 °C and then in increasingly stringent SSC washing solution, with a final wash at 0.1 × SSC at 37 °C. The regional distribution of radioactivity was analyzed in the hippocampal region of wild-type and transgenic sections using a digital autoradiography imager (Micro Imager from Biospace Lab, Paris, France) and its analysis program (BetaVision, Biospace Lab, Paris, France).

### 2.3. Protein Extraction and Western Blot Analysis

HEK293 cells or mouse cortex (pooled from three adult OF1 mice) were homogenized on ice in Tris-buffered saline (100 mM NaCl, 20 mM Tris-HCl, pH 7.4, 1% NP40, 1 × CIP). The homogenates were centrifuged at 13,000× *g* for 10 min at 4 °C and the supernatants were stored at −80 °C. Cell lysate protein concentration was determined using the BCA Protein Assay Kit (ThermoFisher, Les Ulis, France). For SDS-PAGE, 40 μg of protein was diluted in Laemmli 1× (BioRad) with DTT and incubated for 5 mn at 95 °C. Protein samples were loaded in each lane of a 4–15% precast polyacrylamide gel (BioRad, Marnes-la-Coquette, France) and ran in Mini-Protean at 200 V in Tris/Glycine running buffer (BioRad). Following SDS-PAGE, proteins underwent semi-dry electroblotting onto nitrocellulose membranes using the Trans-Blot Turbo Transfer System (BioRad). Membranes were incubated for 1 h at room temperature in blocking solution (PBS 1× containing 5% non-fat dried milk, 0.05% Tween 20) and then overnight at 4 °C with the primary antibody. Membranes were washed in PBS 1× containing 0.05% Tween 20 and incubated for 1 h at room temperature with an anti-mouse, anti-rabbit or anti-goat HRP-conjugated secondary antibody. Membranes were washed three times in PBS 1× containing 0.05% Tween 20. Immune complexes were visualized using the Clarity Western ECL Substrate (BioRad). Chemiluminescence was detected using the ChemiDoc XRS Imaging System (BioRad). As secondary antibodies, we used protein A or protein G IgG, which are HRP-conjugated whole antibodies (1/5000; Abcam ab7460 or ab7456, respectively).

### 2.4. Laser-Assisted Microdissection, Total RNA Preparation and Quantitative Real-Time PCR (Q-RT-PCR) Analysis

Embryonic brain subregions were dissected as shown in Appendix A. The left and right hippocampus was micro-dissected from genotyped P21 mouse brains using a laser-assisted capture microscope (Leica ASLMD instrument) with Leica polyethylene naphthalate membrane slides as described in [20]. RNA preparation and Q-RT-PCR were performed as described in. The Q-RT-PCR results are expressed in an arbitrary unit.

### 2.5. Proximity Ligation Assay (PLA) Analysis

The proximity ligation assay (PLA), also referred to as Duolink^®^ PLA technology (Merck, Lyon, France), detects protein–protein interaction in situ (at distances < 40 nm) at endogenous protein levels [22]. It uses specific antibodies identifying the two proteins of interest and takes advantage of specific DNA primers covalently linked to the antibodies. We followed a protocol similar to that described in [21].

### 2.6. Electrophysiological Analysis on 152F7 Juvenile Mice

The methods used have been described elsewhere [23]. Briefly, for the preparation of hippocampal slices, 21-day-old mice were deeply anesthetized with Nembutal. Brain slices (300–400 µm) were cut in cold artificial cerebrospinal fluid (temperature ≈ 4–8 °C). The artificial cerebrospinal fluid (ACSF) contained 124 mM NaCl, 26 mM NaHCO_3_, 2.5 mM KCl, 1.25 mM NaH_2_PO_4_, 2.5 mM CaCl_2_, 1.3 mM MgCl_2_ and 10 mM glucose. Slices were maintained at room temperature for at least 1 h in a submerged chamber containing ACSF equilibrated with 95% O2 and 5% CO_2_; they were then transferred to a super fusion chamber. Field EPSPs were recorded using microelectrodes (1–3 MΩ) filled with ACSF at 22 to 25 °C. Bipolar stainless-steel electrodes were used for the electrical stimulation of Schaffer collaterals and mossy fibers (0.1 ms, 10 to 100 µA pulses, intertrial intervals of 10 to 30 s). Field EPSP recordings of mossy fibers and Schaffer collaterals were taken with a DAM80 amplifier (WPI), under visual control, with an upright microscope (BX50WI, Olympus, Rungis, France). Mossy fiber LTP was induced by tetanus at 25 Hz for 5 s in the presence of 50 µM D-AP5 (D(-)2-amino-5-phosphonovaleric acid). Mossy fiber responses were identified in wild-type and 152F7 mice with the group 2-metabotropic glutamate receptor selective agonist DCG IV. The inhibitory effects of DCG IV (10 µM) on mossy fiber inputs were similar in mutants and wild-type mice. NBQX (5 µM) was applied at the end of each MF LTP experiment to assess the amplitude of fiber volleys. For Schaffer collateral LTP, stimulating and extracellular recording electrodes were placed in the *stratum radiatum*, and the GABA-A receptor antagonist picrotoxin (100 µM) was added to the ACSF. In this series of experiments, the CA1 region was separated from the CA3 region by sectioning the brain slice with a knife before recording. Data were acquired and analyzed blind to genotype for the LTP experiments. On- and offline data analyses were carried out with Acquis1-ElPhy software (developed by G. Sadoc, UNIC CNRS and ANVAR, France). Summary data are expressed as the means ± SEM. The following drugs were used: NBQX, D-APV, DCGIV (Tocris (Bio-Techne SAS, Noyal Châtillon sur Seiche, France)) and picrotoxin (Sigma (Sigma Aldrich Chimie, Saint-Quentin-Fallavier, France)).

### 2.7. Electrophysiological Analysis of 189N3 Adult Mice

Preparation of Hippocampal Slices 189N3 KI and WT littermate male mice aged 17–22-week-old were shipped from IGBMC (Strasbourg, France) to INMED (Marseille, France). Mice were deeply anesthetized with xylazine 13 mg/kg/ketamine 66 mg/kg and transcardially perfused with a modified artificial cerebrospinal fluid (mACSF) containing the following (in mM) prior to decapitation: 132 choline, 2.5 KCl, 1.25 NaH_2_PO_4_, 25 NaHCO_3_, 7 MgCl_2_, 0.5 CaCl_2_ and 8 D-glucose. The brain was then quickly removed, the hippocampi were dissected and transverse 450 µM thick slices were cut using a Leica VT1200S vibratome in ice-cold oxygenated (95% O_2_ and 5% CO_2_) mACSF. Slices were recovered at room temperature for at least 1 h in artificial cerebrospinal fluid (ACSF) containing the following (in mM): 126 NaCl, 3.5 KCl, 1.25 NaH_2_PO_4_, 26 NaHCO_3_, 1.3 MgCl_2_, 2.0 CaCl_2_ and 10 D-glucose. Both the cutting solution and ACSF were between 290 mOsm and 310 mOsm. All solutions were equilibrated with 95% O_2_ and 5% CO_2_, pH 7.4.

fEPSP Recordings Acute slices were individually transferred to a recording chamber maintained at 30–32 °C and continuously perfused (2 mL/min) with oxygenated ACSF. Field excitatory postsynaptic potential (fEPSP) was recorded in the *lucidum* of CA3 area with glass electrodes (2–3 MΩ; filled with normal ACSF) using a DAM-80 amplifier (low filter, 1 Hz; high pass filter, 3 KHz; World Precision Instruments, Sarasota, FL, USA). Mossy fiber-mediated fEPSPs were evoked by weak electrical stimulations performed via a bipolar NiCh electrode (NI-0.7F, Phymep, Paris, France) positioned in the lucidum of the CA3 area; the stimulus intensity, pulse duration, and frequency were around 30 V, 25 µs, and 0.1 Hz, respectively. Data were digitized with a Digidata 1440 A (Molecular Devices, San Jose, CA, USA) to a PC, and they were acquired using Clampex 10.1 software (PClamp, Molecular Devices). Signals were analyzed offline using Clampfit 10.1 (Molecular Devices). LTP was induced by tetanus at 25 Hz during 5 s; D-AP5 (D-2-amino-5-phosphonovalerate, 40 µM) was included in ACSF during the tetanus to prevent contamination of MF-CA3 LTP with the NMDA receptor-dependent component. At the end of each experiment, 2 µM DCG-IV, a group II mGluR agonist, was bath-applied to confirm the mossy fiber synaptic origin of fEPSP recorded in CA3. The magnitude of long-term plasticity was determined by comparing the baseline-averaged responses before induction with the last 10 min of the experiment. The example traces are the averages of at least 30 consecutive sweeps taken from a single representative experiment.

Statistical analysis All data are shown as the means ±SEM. Statistics were performed using Igor Pro 9 (Wavemetrics, Lake Oswego, OR, USA), and statistical significance was determined by Student’s t test (two-tailed distribution, paired) unless otherwise stated.

### 2.8. Bioinformatics Analysis

Protein–protein networks were analyzed using the STRING bioinformatics suite as in [21]. STRING is a database containing known and predicted protein–protein interactions. The interactions include direct (physical) and indirect (functional) associations; they stem from computational prediction, from the knowledge transfer between organisms, and from interactions aggregated from other (primary) databases.

### 2.9. Reagents

Stock solutions were prepared in water or DMSO, depending on the manufacturers’ recommendation, and stored at −20 °C. Upon experimentation, reagents were bath-applied following dilution into ACSF (1/1000). D-APV and DCGIV were purchased from Tocris Bioscience. Salts for making the cutting solution and ACSF were purchased from Sigma.

### 2.10. Ethics Statement

All experiments were approved by the Institut National de la Santé et de la Recherche Médicale (INSERM) animal care 03882.02 and B751403 agreements (to M Simonneau), in agreement with the European Community Council Directive (2010/63/UE).

The experiments involving 189N3 were approved by the Institut National de la Santé et de la Recherche Médicale (INSERM) animal care and use agreement (D-13-055-19) and the European Community Council Directive (2010/63/UE). The 189N3 animals were produced in an authorized establishment (Institute Clinique de la Souris, license C67-218-40) and in agreement with the European Community Laboratory Animal Care and Use Regulations (2010/63/UE86/609/CEE) and the French Ministry of Agriculture (law 87 848).

## 3. Results

### 3.1. Dyrk1A Up-Regulation Specifically Impairs a Presynaptic Form of LTP at the DG-CA3 Hippocampal Synapse

Two main forms of LTP have been characterized in the mammalian brain. One requires the activation of postsynaptic NMDA (N-methyl d-aspartate) receptors, whereas the other is NMDAR-independent and requires presynaptic mechanisms. The NMDA-dependent LTP at Schaffer Collaterals (SC) of the CA3-CA1 synapse was not modified in 152F7 mice [14]. Furthermore, this SC-CA1 LTP amplitude was only partially decreased in transgenic mice with a *Dyrk1a* trisomy [24]. In contrast, granule cells of dentate gyrus that form the synapse on CA3 neurons via mossy fibers (Figure 1A) [25] display a NMDA-independent form of LTP (MF LTP) [11,12]. MF LTP has been widely characterized and requires the activation of several presynaptic proteins linked to glutamate vesicle exocytosis such as Rims1, Munc13-1, Syn2 and Rab3A [11,12]

Here, we carried out recordings of field excitatory postsynaptic potentials (fEPSPs) in acute hippocampal slices to functionally test the effects of the abnormal up-regulation of DYRK1A in both 152F7 and 189N3 mouse models. MF-mediated fEPSPs were evoked by weak electrical stimulations with a bipolar microelectrode placed in the dentate gyrus. We induced MF LTP by tetanus (see Methods) under the selective NMDA receptor blocker D-AP5 (D-2-amino-5-phosphonovalerate) to prevent NMDA-dependent LTP from occurring (Figure 1B). In 152F7 mice, the increase in the synaptic response of mossy fibers 50 min after tetanus was 116.9 ± 3.7% compared with the baseline response before tetanus. By contrast, in WT mice, the increase in synaptic response 50 min after tetanus was 179.1 ± 5.3%. Thus, MF LTP was deeply impaired in 152F7 mice compared to WT mice (*p* < 0.001, n = 4 mice in both groups; Figure 1C). This LTP was also impaired in 189N3 mice that contain only an extra copy of Dyrk1a (Figure 1D,E). Altogether, these results suggest that Dyrk1A overexpression alone is sufficient to impair MF LTP.

### 3.2. Mouse DYRK1A Interacts with the Chromatin Remodelers EP300 and CREBBP in the Brain

We analyzed the chromatin remodeler protein–protein interaction in order to unravel any possible molecular changes involved in cognition. We focused our analysis on a STRING network composed of DYRK1A, EP300, CREBBP and SMARCA2. This protein-protein network has been validated in humans (https://string-db.org/ (accessed on 21 December 2024)) (Figure 2A). Both EP300 and CREBPP directly interact with SMARCA2, a component of the SWI/SNF complex. The Smarca2-Crebpp interaction was detected by a proximity-dependent biotin identification assay (Intact database) [26] and by an affinity chromatography technology assay (BioGRID database) [27]. The Smarca2–Ep300 interaction was detected by an affinity chromatography technology assay [28].

Here, we proposed the validation of the interactions of these four proteins, DYRK1A, EP300, CREBBP and SMARCA2, in mouse brain. The quantification of these interactions in 189N3 transgenic neurons compared to wild-type neurons indicates a statistically significant increase in DYRK1A-EP300 and DYRK1A-CREBBP interactions (Figure 2B). In contrast, we found no changes in the number of EP300-SMARCA2 and CREBBP-SMARCA2 interactions between 189N3 transgenic neurons compared to wild-type neurons (Figure 2B). Furthermore, the quantitative in situ hybridization (ISH) using the Allen Brain Atlas indicates that *Dyrk1a*, *Ep300* and *Crebbp* are highly expressed in adult mouse hippocampi (Figure 2C). Altogether, these results suggest that this increase in the number of DYRK1A-EP300 and DYRK1A-CREBBP interactions in the 189N3 brain can induce changes in the transcriptome of hippocampal neurons.

### 3.3. Dyrk1A Up-Regulation Affects the Transcription of Genes Encoding Presynaptic Proteins

We next studied the possible deregulation of genes involved in synaptic plasticity. As NMDA-independent LTP depends on presynaptic proteins and was found to be affected in 152F7 and 189N3 mice (this study) (Figure 1), we focused our analysis on genes encoding presynaptic proteins and involved in the NMDA-independent LTP.

The MF-CA3 synapse [25] display a NMDA-independent form of LTP [11,12]. This presynaptic form of LTP [11,12] has been widely characterized and requires the activation of several presynaptic proteins linked to glutamate vesicle exocytosis such as RIMS1, MUNC13-1, SYN2 and RAB3A [29,30]. *Rims1, Munc13-1* and *Rab3a* knockout transgenic lines display impaired NMDA-independent LTPs [31,32,33,34]. In contrast, both NMDA -dependent and -independent long-term potentiations are not affected in mice lacking synapsins [35].

RIMS1, MUNC13-1, SYN2 and RAB3A are part of a functional network as evidenced by the STRING bioinformatics suite, and they are members of a GO:0017156 Calcium-ion regulated exocytosis ensemble (false discovery rate: 4.89 × 10^−7^) [36] (Appendix A). We used RT-Q-PCR, following the laser-assisted micro-dissection of the hippocampal sub-regions, and dentate gyrus, CA3 and CA1 regions, respectively (Figure 3B), to quantify any possible changes in transcripts between 152F7 and control mice. We found *that Rims1, Syn2, Munc13-1* and *Rab3a* levels significantly decreased in the 152F7 dentate gyrus and CA3 compared to the wild type (Figure 3C). We confirmed the decrease in *Rims1* transcripts using quantitative radioactive ISH on hippocampal sections of 152F7 and wild-type (WT) mice (Figure 3D,E). Altogether, these results show that changes in *Dyrk1a* dosage can modify the expression of gene-encoding proteins involved in the molecular mechanism of presynaptic vesicle release.

## 4. Discussion

The general aim of this study was to demonstrate a relationship between the cognitive defects and synapse plasticity impairments in the 152F7 Down syndrome mouse model [14], in which episodic spatial memory is impaired without noticeable defects in NMDA-dependent LTP at the SC-CA1 synapse [14,16].

The choice of the appropriate mouse model for Down syndrome is an important issue [37,38]. A previous mouse model, known as Ts65Dn, has been considered the standard for Down syndrome research, used in preclinical studies for nearly 30 years. However, the Ts65Dn mouse genome contains 45 extra genes that are irrelevant to human Down syndrome, limiting the translation of the mechanisms identified in mouse to human Down syndrome [37,39]. We studied hippocampal NMDA-independent synaptic plasticity [16,17] in both 152F7 and 189N3 mice. The 152F7 mouse model expresses a human YAC fragment of 570 kb that includes six distinct genes, among which two are transcription regulators, RIPPLY3 and DYRK1A. RIPPLY3 was characterized as a transcriptional co-repressor known to be regulating TBX1, a major gene involved in DiGeorge syndrome. A recent study has shown that RIPPLY3 overdosage in DS models is important for the mid-face shortening observed in the cranium of models which mimic human phenotypes [40]. In addition, RIPLY3 overdosage could lead to TBX1 down-regulation in other organs, such as the brain, during development where it could interact with DYRK1A overdosage as it interacts with DYRK1A during DS model branchial arch morphogenesis. In contrast, the 189N3 line includes only one extra copy of *DYRK1A*. In line 152F7, we found that NMDA-independent LTP was completely suppressed. In 189N3 mice, the NMDA-independent LTP was impaired. The involvement of DYRK1A in plasticity-related processes, including LTP between CA3 and CA1 neurons, has been previously reported in the Ts65Dn mouse model [41,42,43].

Altogether, these results indicate the involvement of DYRK1A overexpression in NMDA-independent LTP impairment. Other gene deregulation may also be involved, explaining the complete suppression of the presynaptic LTP in the 152F7 line. We can hypothesize that the triplication of RIPPLY3 may contribute to this, as this gene is highly expressed in the mouse hippocampus (see Allen Brain Atlas data; https://mouse.brain-map.org/gene/show/82067 (accessed on 21 December 2024)). Similarly, NMDA-dependent LTP can be rescued either by the copy number restauration of Dyrk1a levels [17] or Brwd1 [44] in the Ts65Dn mouse model of Down syndrome.

It was shown that DYRK1A regulates the transcription of a subset of genes by its association to their proximal promoter regions and by phosphorylating the C-terminal domain of the RNA polymerase II [45]. Furthermore, DYRK1A interacts with histone acetyl transferase p300 and CBP and localizes to enhancers [46]. In this study, we found that DYRK1A directly interacts with chromatin remodelers EP300 and CREBBP that are known to impact synaptic plasticity [47,48,49]. EP300 and CREBPP are two members of the p300-CBP coactivator family containing a histone acetyltransferase (HAT) domain involved in chromatin remodeling, and mutations in these genes have been shown to cause Rubinstein–Taybi syndrome [50].

From these results, we examined whether *DYRK1A* gene dosage can modify the regulation of genes that encode proteins involved in presynaptic vesicle exocytosis as the knockout of the related genes disrupts MF-CA3 non-NMDA LTP. Using quantitative ISH, we were able to find a statistically significant decrease in Rims1 in 152F7 dentate gyrus. We identified a significant decrease in transcript levels involved in presynaptic vesicle exocytosis such as those of *Rims1, Rab3a* and *Munc13-1*, whose knockout induces impairment in MF-CA3 LTP. Other transcripts, such as *Syn2*, involved in presynaptic functioning were also impacted. Impairment of MF-CA3 LTP has been reported only for the knockouts of *Rims1, Rab3* and *Munc13-1*, with no effect on haploinsufficiency. Here, we propose that the combined haploinsufficiency for *Rims1*, *Rab3* and *Munc13-1* can have a functional consequence similar to the knockout of one of these genes, leading to a phenocopy of the knockout. Alternatively, DYRK1A can also directly phosphorylate SYN1, which controls the reserve synaptic pool as demonstrated in the DS models [18].

An important advance in the improvement of cognition in mouse models of Down syndrome and Alzheimer disease, as well as in human Down syndrome, was recently reported [41]. Gonadotropin-releasing hormone (GnRH) was shown to improve cognition when GnRH levels were restored in brain. In the Ts65Dn mouse model, GnRH restoration rescues hippocampal transcriptome and connectivity [4]. Analysis of NMDA-independent LTP in 152F7 and 189N3 mouse models with manipulation of GnRH levels would be instrumental to deciphering the links between GnRH targets and synaptic plasticity.

This study has limitations. The selective copy number restoration of *Dyrk1a* in trisomic animals is needed to demonstrate *Dyrk1a* gene dosage involvement in NMDA-independent LTP. Furthermore, emerging spatial transcriptomics methods may allow us to characterize molecular profiles of hippocampal cell types with a spatial resolution [51].

Altogether, this work identifies that NMDA-independent LTP is impaired in mouse models of DYRK1A up-regulation, involving DYRK1A-linked epigenetics mechanisms that impact the expression of genes encoding presynaptic proteins, and defines a novel endophenotype. This study thereby provides a novel mechanistic and potentially therapeutic understanding of deregulated signaling downstream of DYRK1A up-regulation.

## Figures and Tables

**Figure 1 life-15-00149-f001:**
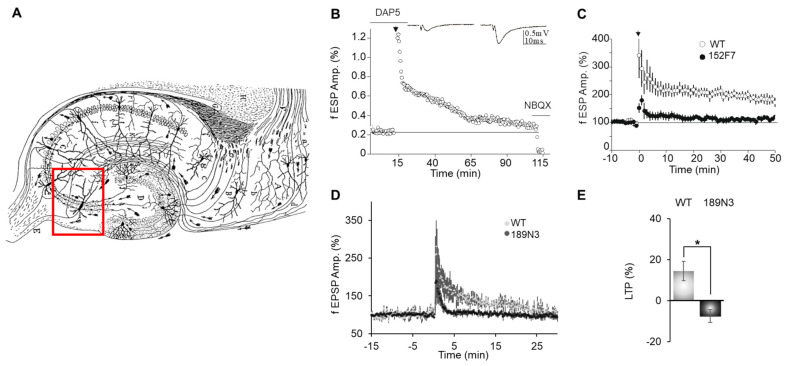
Presynaptic LTP between dentate gyrus mossy fibers and CA3 impaired in both adult 152F7 and 189N3 mouse hippocampi as compared to control mice. (**A**). Illustration of a hippocampal (Ramón y Cajal) sagittal slice. The red box indicates a CA3 pyramidal neuron that receives mossy fibers. In the hippocampus, the granule cells of the dentate gyrus give rise to mossy fiber axons, which travel into the CA region. Wikipedia file (https://en.m.wikipedia.org/wiki/File:CajalHippocampus.jpeg (accessed on 21 December 2024)). (**B**). Time course of mossy fiber LTP in a wild-type mouse. Data are expressed as the means ± SEM and calculated from 4 different mice with 3 slices per mouse for both genotypes. Mossy fiber LTP was induced by a single tetanus of 25 Hz (for 5 s, black arrow) in the presence of 50 µM of DAP5. NBQX was applied to obtain information concerning the fiber volley. (**C**). Time course (mean ± s.e.m.) of mossy fiber LTP in transgenic 152F7 compared to WT mice. In 152F7 mice, the increase in the synaptic response of mossy fibers 50 min after tetanus was 116.9 ± 3.7% compared with the baseline response before tetanus. By contrast, in WT mice, the increase in synaptic response 50 min after tetanus was 179.1 ± 5.3%. Thus, MF LTP was deeply impaired in 152F7 mice compared to WT mice (*p* < 0.001, n = 4 mice in both groups). The black arrow indicates the peak of the LTP. (**D**). Time course (mean ± s.e.m.) of mossy fiber LTP in transgenic 189N3 compared to WT mice. (**E**). Distribution for the magnitude of LTP observed in WT and189N3 and mice. The magnitude of long-term plasticity was determined by comparing baseline-averaged responses before induction with the last 10 min of the experiment. This magnitude was impaired with (mean ± s.e.m.) WT 14.4 ± 4.7 % and 189N3 −7.4 ± 3.8 % (n = 4 for each genotype). * *p* < 0.01. Each group of 4 mice included 2 females and 2 males.

**Figure 2 life-15-00149-f002:**
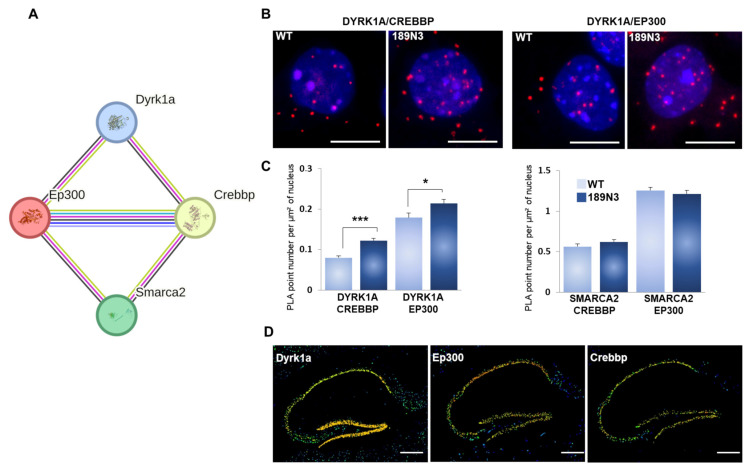
Interactions of Dyrk1a with chromatin remodelers. (**A**). Schematic representation of Dyrk1a interactions with EP300 and CREBBP. (**B**). In situ proximity ligation assays (PLA) on primary cortical neurons fixed at DIC7 (red fluorescence) using anti-Dyrk1a and anti-Ep300; anti-CREBBP, anti-SMARCA2 and anti-EP300; or anti-CREBBP antibodies. (**C**). Nuclear bodies were labeled using Topro3 staining (blue fluorescence). The mean interaction point numbers were calculated in a nuclear body of 45 to 89 cortical neurons at DIC7 (from 3 to 5 different embryos per genotype). PLA using anti-Ep300 and anti-Fibrillarin antibodies were performed as a negative control and no difference was shown between transgenic 189N3 and WT cortical neurons. Scale bars = 10 μm. * *p* < 0.05; *** *p* < 0.0005. (**D**). False-color image of ISH from the Allen Brain Atlas showing *Dyrk1a*, *Ep300* and *Crebbp* transcript expressions in an adult mouse hippocampus. The red arrows arrow heads indicate the expected position of the three distinct proteins in the gels. Scale bar = 100 µm.

**Figure 3 life-15-00149-f003:**
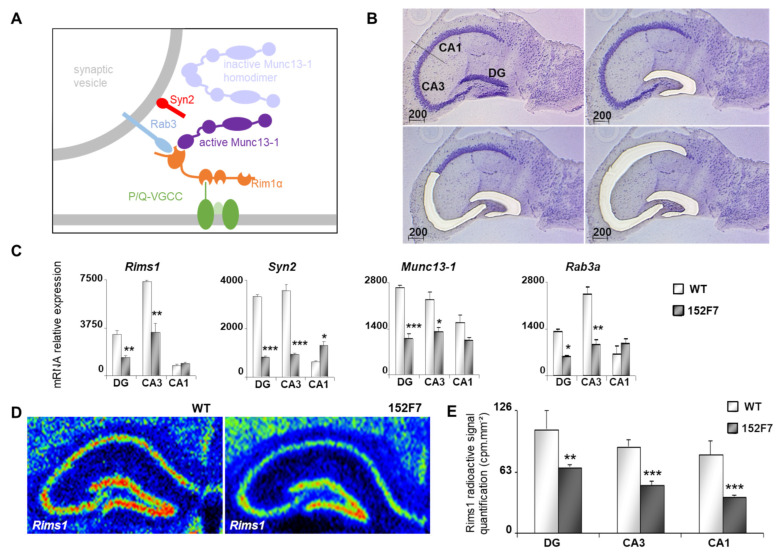
Presynaptic protein expression in the adult 152F7 mouse hippocampus as compared to control mice. (**A**). Schematic representation of molecules RIMS1, SYN2, RAB3A and MUNC13A involved in the glutamate release from presynaptic vesicles. (**B**)**.** Laser-assisted microdissection of the three subregions of the P21 mouse hippocampus stained with toluidine blue. Scale bar = 200 µm. (**C**). From the laser-assisted microdissection of the three subregions of the P21 mouse hippocampus, *Rims1, Syn2, Rab3a* and *Munc13-1* transcripts are shown to be down-regulated in the DG and CA3 hippocampal subregions of juvenile transgenic 152F7 mice compared to their WT siblings, as shown by the Q-RT-PCR analysis. Rims1: (mean ± s.e.m.) WT DG level 3221 ± 300, 152F7 DG level 1422 ± 161; WT CA3 level 7366 ± 83, 152F7 CA3 level 3349 ± 83; WT CA1 level 806 ± 89, 152F7 CA1 level 978 ± 100; *Syn2*: (mean ± s.e.m.) WT DG level 20020 ± 457, 152F7 DG level 4950 ± 360; WT CA3 level 21520 ± 1464, 152F7 CA3 level 5647 ± 248; WT CA1 level 3746 ± 320, 152F7 CA1 level 7709 ± 1028; Munc13-1: (mean ± s.e.m.) WT DG level 2645 ± 68, 152F7 DG level 1097 ± 161; WT CA3 level 2279 ± 234, 152F7 CA3 level 1313 ± 125; WT CA1 level 1587 ± 245, 152F7 CA1 level 1049 ± 72; Rab3a: (mean ± s.e.m.) WT DG level 1338 ± 174, 152F7 DG level 566 ± 34; WT CA3 level 2440 ± 219, 152F7 CA3 level 947 ± 136; WT CA1 level 658 ± 96, 152F7 CA1 level 961 ± 144; (n = 3) Scale bar = 1 mm. * *p* < 0.01 ** *p* < 0.001 *** *p* < 0.0001. (**D**). Quantification of Rims1 RNA in WT and 152F7 mouse hippocampi using Quantitative In Situ Hybridation (Q-ISH). False-color image of antisense Rims1 RNA Q-ISH of hippocampi from juvenile P21 WT and 152F7 mice. Q-ISH was performed using 3H radioactive probes for Rims1. (**E**). Q-ISH indicates a significant down-regulation of *Rims1* in the three subregions of the 152F7 mouse hippocampus. Rims1 (mean ± s.e.m.) WT DG level 107 ± 19, 152F7 DG level 68 ± 3; WT CA3 level 89 ± 7, 152F7 CA3 level 50 ± 3; WT CA1 level 80 ± 14, 152F7 CA1 level 38 ± 1; (n = 10 for WT; n = 20 for 152F7). Scale bar = 1 mm. ** *p* < 0.001, *** *p* < 0.0001.

## Data Availability

The data presented in this study are available on request from the corresponding author. These data are raw data used to generate the Figures.

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
