# Peer review of "DYRK1A Up-Regulation Specifically Impairs a Presynaptic Form of Long-Term Potentiation"

_life, 2025, doi:10.3390/life15020149_

Round 1
Reviewer 1 Report
Comments and Suggestions for Authors
The authors utilized 152F7 mice and demonstrated that a presynaptic form of NMDA-independent LTP at hippocampal mossy fibers was impaired in this mouse model. They further analyzed Dyrk1a interactions with chromatin remodelers and found that the number of DYRK1A-EP300 and DYRK1A-CREBPP complexes increased in 152F7 mice. Additionally, they used 189N3 mice and observed a similar impairment in NMDA-independent LTP. Their findings indicate that up-regulation of Dyrk1a is sufficient to specifically inhibit NMDA-independent LTP, and they suggested that this inhibition is associated with chromatin changes that deregulate the expression of genes encoding proteins involved in glutamate synaptic release. This topic is both interesting and innovative. However, several issues need to be addressed:
1: Introduction is poorly described. This section should focus on providing the background, research objectives and research significance of the study. Lines 77–96, which discuss work related to this study, should be consolidated into a single paragraph and described more concisely. Furthermore, the background context of the study should be elaborated.
2: The authors used two types of mouse models, 152F7 and 189N3, to model Down syndrome. However, they only described deficits in memory and learning observed in these mice within behavioral models. What other phenotypes of Down syndrome are present in these mouse models? Why were these specific models chosen to represent Down syndrome? Is there sufficient literature to support their use?
3: In Results section 3.2 (Lines 273–284), the authors introduced some background and prior research findings. Could these foundational studies be moved to the Introduction section? The Results section should primarily focus on the findings of this study. If necessary, only a brief rationale for the experiments and the key conclusions being tested should remain in this section.
4: In Results section 3.2, the authors mention that data from the Allen Brain Atlas indicates that Dyrk1a, Ep300, and Crebbp are highly expressed in the adult mouse hippocampus. Do the authors have their own hippocampal data from mice to corroborate these findings and show the expression of Dyrk1a, Ep300, and Crebbp?
5: In hippocampal slices from mice, are there results to confirm the interaction between DYRK1A-EP300 and DYRK1A-CREBBP? Can conclusions similar to those shown in Fig. 2B be drawn from these experiments?
6: The y-axes in the bar graphs of Fig. 2 and Fig. 3 lack unit labels. Please provide appropriate unit annotations.

Author Response
The 152F7 mouse model incorporates 570 kb fragment of Hsa21 with six protein-coding genes including DYRK1A (Supplementary Figure 1). One of the six genes, DSCR9 is a primate-specific gene (15). The syntenic region in mouse chromosome 16 genome is more condensed with ~350 kb (Supplementary Figure 2) instead of ~570 kb in Hsa21. The organization of genes is similar between human and mouse genome with the sequence 3’ to 5’: Ripply3, Pigp, TTC3, Vps26c and Dyrk1a.
4: In Results section 3.2, the authors mention that data from the Allen Brain Atlas indicates that Dyrk1a, Ep300, and Crebbp are highly expressed in the adult mouse hippocampus. Do the authors have their own hippocampal data from mice to corroborate these findings and show the expression of Dyrk1a, Ep300, and Crebbp?
Allen Brain data are widely used in many important papers and considered as bona fide reference data. It was recently showed that multiplexed error-robust fluorescence in situ hybridization (MERFISH), a spatially resolved single-cell transcriptomics method, able to generate a comprehensive, molecularly defined and spatially resolved cell atlas of the entire adult mouse brain. showed excellent agreement with the Allen Brain Atlas in situ hybridization data (ref Zhang et al., 2023, Nature).
Zhang M, Pan X, Jung W, Halpern AR, Eichhorn SW, Lei Z, Cohen L, Smith KA, Tasic B, Yao Z, Zeng H, Zhuang X. Molecularly defined and spatially resolved cell atlas of the whole mouse brain. Nature. 2023 Dec;624(7991):343-354. doi: 10.1038/s41586-023-06808-9.
We did not repeat ISH experiments for Dyrk1a, Ep300, and Crebbp genes.
5: In hippocampal slices from mice, are there results to confirm the interaction between DYRK1A-EP300 and DYRK1A-CREBBP? Can conclusions similar to those shown in Fig. 2B be drawn from these experiments?
PLA was done on cultured mouse neurons. The sensibility of the PLA analysis is higher in cultures than in hippocampal sections. This is why we limited our analysis to primary neuronal cultures.
6: The y-axes in the bar graphs of Fig. 2 and Fig. 3 lack unit labels. Please provide appropriate unit annotations.
This point was corrected in Fig.2 and Fig.3, accordingly.
Reviewer 2 Report
Comments and Suggestions for Authors
The study explores a very interesting topic, i.e., the involvement of DYRK1A up-regulation in the presynaptic type of long-term potentiation, based on preclinical data, with potential theoretical and practical benefits for research in the field of Down syndrome, Alzheimer's dementia, and Autism Spectrum Disorder. However, the preprint version of this manuscript is more than 5 years and a half older than the current manuscript, and more than 2/3 of the text is unchanged; could the Author explain this delay and if other studies were performed in the meantime that could have relevance on this topic? There are some other articles related to the involvement of DYRK1A in plasticity-related processes that were not mentioned- https://pubmed.ncbi.nlm.nih.gov/31803016/, about the DYRK1A and mitochondrial functioning- https://pubmed.ncbi.nlm.nih.gov/36590914/, or about DYRK1A and LTP, specifically- https://pubmed.ncbi.nlm.nih.gov/37997361/.
Abstract- the acronym “NMDA”, i.e., N-methyl-D-aspartate, is not defined either here nor in the body text;
Lines 41-42- keywords are not listed
Lines87-92- a reference is needed here;
Lines 100-102- Why are these digital object identifiers mentioned here? The reference numbers should be enough to identify the papers; also, please check the second “doi” because it does not correspond to reference (10) mentioned;
Line 132- Supplementary Table S6 -where is this table presented because, in the “Supplementary materials” folders, there are only figures.
Line 138- Supplementary Table S3- same question as above.
Line 143- Which Supplementary Figure are the Authors referring to?
Ines 391-393- maybe the Author should reconsider calling “recent”, the article published by di Vona et al. in 2015;
Lines 423-427—Based on the limitations of the current study or controversies in the literature, are there any directions suggested for further research in this field?
Lines 444-478 and 481-488 are remnants of the journal’s template, and they include important sections that the Authors must complete.
References- (36) is incomplete.
Author Response
The study explores a very interesting topic, i.e., the involvement of DYRK1A up-regulation in the presynaptic type of long-term potentiation, based on preclinical data, with potential theoretical and practical benefits for research in the field of Down syndrome, Alzheimer's dementia, and Autism Spectrum Disorder.
1-However, the preprint version of this manuscript is more than 5 years and a half older than the current manuscript, and more than 2/3 of the text is unchanged; could the Author explain this delay and if other studies were performed in the meantime that could have relevance on this topic? There are some other articles related to the involvement of DYRK1A in plasticity-related processes that were not mentioned- https://pubmed.ncbi.nlm.nih.gov/31803016/, about the DYRK1A and mitochondrial functioning- https://pubmed.ncbi.nlm.nih.gov/36590914/, or about DYRK1A and LTP, specifically- https://pubmed.ncbi.nlm.nih.gov/37997361/.
The delay in submission of was linked to complementary studies that were complicated by the Covid period that needed animal house facilitities.
We now mention the three articles related to the involvement of DYRK1A in plasticity-related processes (New text lines 387-390)
The involvement of DYRK1A in plasticity-related processes including LTP between CA3 and CA1 neurons has been previously reported in Ts65Dn mouse model (43–45).
2-Abstract- the acronym “NMDA”, i.e., N-methyl-D-aspartate, is not defined either here nor in the body text;
N-methyl-D-aspartate is now indicated in the abstract (New text line 25)
3-Lines 41-42- keywords are not listed
Down Syndrome, Synapse, Long-term potentiation, NMDA-independent LTP, Pre-synaptic mechanisms, Epigenetics (New text lines 39-40)
4-Lines 87-92- a reference is needed here;
These lines were rewritten (New text lines 94-98)
To get insights on molecular mechanisms involved in such synaptic changes, we analyzed DYRK1A interactions with chromatin remodelers. We evidenced that the number of Dyrk1a-Ep300 and Dyrka-Crebbp increased in 152F7 mice, using proximity ligation assay (PLA) technology. Moreover, we observed a transcriptional decrease of genes encoding presynaptic proteins involved in glutamate vesicle exocytosis, namely Rims1, Munc13-1, Syn2 and Rab3a.
was modified as
Here, we evidenced that the number of Dyrk1a-Ep300 and Dyrka-Crebbp increased in 152F7 mice, using proximity ligation assay (PLA) technology. Moreover, we found a transcriptional decrease of genes encoding presynaptic proteins involved in glutamate vesicle exocytosis, namely Rims1, Munc13-1, Syn2 and Rab3a.
5-Lines 100-102- Why are these digital object identifiers mentioned here? The reference numbers should be enough to identify the papers; also, please check the second “doi” because it does not correspond to reference (10) mentioned;
Lines 100-102 were corrected
6-Line 132- Supplementary Table S6 -where is this table presented because, in the “Supplementary materials” folders, there are only figures.
Primary antibodies used were as shown in Supplementary Table S6.
This point was corrected.
7-Line 138- Supplementary Table S3- same question as above.
Primary antibodies used were as shown in Supplementary Table S3.
This point was corrected.
8-Line 143- Which Supplementary Figure are the Authors referring to?
Embryonic brain subregions were dissected as shown in Supplementary Figure.
This line was suppressed.
9-Iines 391-393- maybe the Author should reconsider calling “recent”, the article published by di Vona et al. in 2015;
It was recently shown that the DYRK1A regulates transcription of a subset of genes
The “recently” was removed.
10-Lines 423-427—Based on the limitations of the current study or controversies in the literature, are there any directions suggested for further research in this field?
According to comment of the reviewer 3, we present more precisely our results.
Altogether, this work identifies novel molecular and functional novel biomarkers that are involved in the cognitive signature displayed by mouse models of Down Syndrome. Our study thereby provides a novel mechanistic and potentially therapeutic understanding of deregulated signaling downstream of DYRK1A up-regulation.
Was modified by (Next text lines 431-435)
Altogether, this work identifies that NMDA-independent LTP is impaired in mouse models of DYRK1A up-regulation, involving DYRK1A-linked epigenetics mechanisms that impact the expression of genes encoding presynaptic proteins and defines a novel endophenotype. This study thereby provides a novel mechanistic and potentially therapeutic understanding of deregulated signaling downstream of DYRK1A up-regulation.
11-Lines 444-478 and 481-488 are remnants of the journal’s template, and they include important sections that the Authors must complete.
Author Contributions: For research articles with several authors, a short paragraph specifying their individual contributions must be provided. The following statements should be used “Conceptualization, X.X. and Y.Y.; methodology, X.X.; software, X.X.; validation, X.X., Y.Y. and Z.Z.; formal analysis, X.X.; investigation, X.X.; resources, X.X.; data curation, X.X.; writing—original draft preparation, X.X.; writing—review and editing, X.X.; visualization, X.X.; supervision, X.X.; project administration, X.X.; funding acquisition, Y.Y. All authors have read and agreed to the published version of the manuscript.” Please turn to the CRediT taxonomy for the term explanation. Authorship must be limited to those who have contributed substantially to the work reported.
We indicated:
Author Contributions: Conceptualization, AM.LB. and M.S.; investigation, AM.LB., S.H., J.V., P.S., A.D.; resources, Y.H.; writing—original draft preparation, M.S.; writing—review and editing, AM.LB., M.S.; supervision, M.S.; project administration, M.S.; funding acquisition, M.S. All authors have read and agreed to the published version of the manuscript.
Conflicts of Interest: Declare conflicts of interest or state “The authors declare no conflict of interest.” Authors must identify and declare any personal circumstances or interest that may be perceived as inappropriately influencing the representation or interpretation of reported research results. Any role of the funders in the design of the study; in the collection, analyses or interpretation of data; in the writing of the manuscript, or in the decision to publish the results must be declared in this section. If there is no role, please state “The funders had no role in the design of the study; in the collection, analyses, or interpretation of data; in the writing of the manuscript, or in the decision to publish the results”.
We indicated:
Conflicts of Interest: The authors declare no conflict of interest. The funders had no role in the design of the study; in the collection, analyses, or interpretation of data; in the writing of the manuscript, or in the decision to publish the results.
12-References- (36) is incomplete.
Vona CD, Bezdan D, Islam ABMMK, Salichs E, López-Bigas N, Ossowski S, et al. Chromatin-wide profiling of DYRK1A reveals a role as a gene-specific RNA polymerase II CTD kinase. Molecular Cell. 2015;
The reference was completed as:
Vona CD, Bezdan D, Islam ABMMK, Salichs E, López-Bigas N, Ossowski S, et al. Chromatin-wide profiling of DYRK1A reveals a role as a gene-specific RNA polymerase II CTD kinase. Molecular Cell. 2015 Feb;57(3):506-20.
Reviewer 3 Report
Comments and Suggestions for Authors
The authors are using 2 mouse models of Down syndrome (DS): the Tg (CEPHY152F7)12Hgc line(noted here 152F7 line) that displays a ~570 kb human genomic region surrounding the DYRK1A gene that includes five other genes; a second line, the Dyrk1A BAC model (Tg(Dyrk1a)189N3Yah, noted here 189N3 line) that gives a triplication of ~152 kb mouse Dyrk1a locus. Both models show an overexpression of human DYRK1A and mouse Dyrk1a and represent a valuable tool to investigate the molecular mechanisms of cognitive impairment in DS.
The authors have investigated Long-term potentiation (LTP) at the Dentate Gyrus granule cells tp CA3 pyramidal cell synapses that is known to be a NMDA-independent presynaptic LTP and linked to presynaptic proteins, using extracellular field recording in hippocampal slices.
Proximity ligation assay (PLA) is not mentioned in the method section.
Results
Figure 1. Please indicate if N-4 were male or female animals for each phenotype.
Lines 279-280. The authors are reporting previous published results (ref 17, 18) in the result section. This paragraph is also mentioned in the introduction, and it should be better included in the discussion or deleted here.
Lines 294-298. Here again the authors are reporting previous results from the same study (Ref 18).
Figure 2. The quality of Fig 2B for PLA is low. Could the authors improve the image resolution.
The last paragraph of the discussion Lines 424-427 is too general, and the authors should clarify which is the main novel finding in Dyrk1a regulation.
Minor comments
Abstract line 24 …Down syndrome should not be defined as a psychiatric disorder, better a systemic genetic condition.
Section 2.7 is too short and the authors should expand it.
Author Response
The authors are using 2 mouse models of Down syndrome (DS): the Tg (CEPHY152F7)12Hgc line(noted here 152F7 line) that displays a ~570 kb human genomic region surrounding the DYRK1A gene that includes five other genes; a second line, the Dyrk1A BAC model (Tg(Dyrk1a)189N3Yah, noted here 189N3 line) that gives a triplication of ~152 kb mouse Dyrk1a locus. Both models show an overexpression of human DYRK1A and mouse Dyrk1a and represent a valuable tool to investigate the molecular mechanisms of cognitive impairment in DS.
The authors have investigated Long-term potentiation (LTP) at the Dentate Gyrus granule cells tp CA3 pyramidal cell synapses that is known to be a NMDA-independent presynaptic LTP and linked to presynaptic proteins, using extracellular field recording in hippocampal slices.
Proximity ligation assay (PLA) is not mentioned in the method section.
We added PLA analysis : new lines 152-157
2.5. Proximity Ligation Assay (PLA) Analysis
Proximity ligation assay (PLA), also referred to as Duolink® PLA technology, detects protein-protein interactions in situ (at distances < 40 nm) at endogenous protein levels (21). It uses specific antibodies identifying the two proteins of interest and takes advantage of specific DNA primers covalently linked to the antibodies. We followed a protocol similar to that described in (20).
Results
Figure 1. Please indicate if N-4 were male or female animals for each phenotype.
We added: Each group of 4 mice included 2 females and 2 males (new text line 285-286).
Lines 279-280. The authors are reporting previous published results (ref 17, 18) in the result section. This paragraph is also mentioned in the introduction, and it should be better included in the discussion or deleted here.
We previously found that DYRK1A interacts with the REST/NRSF-SWI/SNF chromatin remodeling complex (17) This deregulation induces changes in expression of gene clusters involved in the neuronal phenotypic traits of Down syndrome (17). Furthermore, by combining analyses of exome sequencing and mass spectroscopy in these mouse models, we also found uncovered two deregulated repertoires associated with chromatin and synaptic pathways (18).
This paragraph is now deleted.
Lines 294-298. Here again the authors are reporting previous results from the same study (Ref 18).
Using a high-throughput, domain-based yeast two-hybrid (Y2H) technology against a human brain library (18), we identified direct interactions between DYRK1A and EP300 or CREBBP (18). We validated these interactions by immunoprecipitation (IP) in HEK293 cells using anti-EP300 and anti-CREBBP antibodies and successfully identified DYRK1A (18).
This sentence is now deleted.
Figure 2. The quality of Fig 2B for PLA is low. Could the authors improve the image resolution.
Resolution of Fig. 2B was improved.
The last paragraph of the discussion Lines 424-427 is too general, and the authors should clarify which is the main novel finding in Dyrk1a regulation.
Altogether, this work identifies novel molecular and functional novel biomarkers that are involved in the cognitive signature displayed by mouse models of Down Syndrome. Our study thereby provides a novel mechanistic and potentially therapeutic understanding of deregulated signaling downstream of DYRK1A up-regulation.
Was modified by: (new text lines 431-435)
Altogether, this work identifies that NMDA-independent LTP is impaired in mouse models of DYRK1A up-regulation, involving DYRK1A-linked epigenetics mechanisms that impact the expression of genes encoding presynaptic proteins and defines a novel endophenotype. This study thereby provides a novel mechanistic and potentially therapeutic understanding of deregulated signaling downstream of DYRK1A up-regulation.
Minor comments
Abstract line 24 …Down syndrome should not be defined as a psychiatric disorder, better a systemic genetic condition.
Chromosome 21 DYRK1A kinase is associated with a variety of psychiatric diseases including Down Syndrome.
We now indicated: (new text line 21)
Chromosome 21 DYRK1A kinase is associated with a variety of neuronal diseases including Down Syndrome.
We want to indicate that mutations in DYRK1A gene are found in Autism spectrum Disorders (see Stessman et al., 2017; Levy et al., 2021) and that three copies of DYRK1A gene are found in Down syndrome.
Stessman et al.. Targeted sequencing identifies 91 neurodevelopmental-disorder risk genes with autism and developmental-disability biases. Nat Genet. 2017 Apr;49(4):515-526. doi: 10.1038/ng.3792.
Levy JA, LaFlamme CW, Tsaprailis G, Crynen G, Page DT. Dyrk1a Mutations Cause Undergrowth of Cortical Pyramidal Neurons via Dysregulated Growth Factor Signaling. Biol Psychiatry. 2021 Sep 1;90(5):295-306. doi: 10.1016/j.biopsych.2021.01.012.
Section 2.7 is too short and the authors should expand it.
Protein-Protein networks were analyzed using String bioinformatics suite as in (18).
We added: (new text lines 219-224)
Protein-Protein networks were analyzed using String bioinformatics suite as in (20). STRING is a database of known and predicted protein-protein interactions. The interactions include direct (physical) and indirect (functional) associations; they stem from computational prediction, from knowledge transfer between organisms, and from interactions aggregated from other (primary) databases.
Round 2
Reviewer 1 Report
Comments and Suggestions for Authors
I have no further questions.
Reviewer 2 Report
Comments and Suggestions for Authors
The quality of the manuscript improved
Reviewer 3 Report
Comments and Suggestions for Authors
The authors have responded to all my concerns